# Topic Aware Transformer: Domain Shift for Unconditional Text Generation Models

## Abstract

Our goal is to guide pre-trained language models (PLMs) towards unconditional text generation tasks while resolving the domain gap and avoiding the catastrophic forgetting. Because Transformer-based models are pretrained on more massive and heterogeneous corpora than specific target corpus, the gap between these corpora and the target corpus raises the question of whether these PLMs will actually benefit this task even after fine-tuning. As the domain adaptation of PLMs needs to bridge this gap, we propose a framework, Topic Aware Transformer (TAT), that adapts PLMs for target-aware text generation while alleviating catastrophic forgetting. The motivation of TAT to distill the target-specific knowledge as topics, and steer PLMs toward these topics. This requirement and motivation lead us to introduce a topic steering layer (TSL) as an additional layer, and Topic Distribution Modeling (TDM) as a training task. Experiments show that these components resolve the gap as the domain shift, and can tailor PLMs to generate text to better reflect a given small fine-tuning corpus.

## 1 INTRODUCTION

Our goal is to adapt pre-trained language models (PLMs) to achieve unconditional text generation toward a target domain. The success of Transformer-based PLMs motivates us to explore how to fine-tune them so as to well reflect a given target corpus thereby generating more personalized texts with very few specializations. The size of the target corpus is generally much smaller than that of existing pre-training corpora, which may lead to catastrophic forgetting (Ramasesh et al., 2021). For example, the popular pre-training data sets of Giga5en (Parker et al., 2011), and ClueWeb 2012-B[1] occupy 16G, and 25TB, respectively. PLMs can become biased toward the patterns of language used in the training data (Keskar et al., 2019). Given the rapid diversification of applications, a pre-training approach is needed to effectively achieve domain shift without catastrophic forgetting.

Toward this domain shift, we propose a framework, Topic Aware Transformer (TAT), that adapts PLMs as unconditional generative tasks while alleviating catastrophic forgetting. As the domain knowledge consists of global (e.g., linguistic) and specific (e.g., semantic) knowledge, our intuition is that knowledge can be represented as a distribution of words, and the gap between the source and the target domain can be taken to be differences between distributions. These intuitions motivate TAT to detect these distributions via topics, and steer PLMs toward these topics to highlight the target-specific knowledge. That is, the motivation of TAT is to introduce a topic steering layer (TSL) as an additional layer that detects topics and helps training PLMs theoretically, and Topic Distribution Modeling (TDM) as a training task to align text on the topic representation on the target domain. To prevent catastrophic forgetting, TAT can fine-tune PLMs while bridging the domain gap without updating PLM parameters.

Experiments confirm that TAT supports PLMs and verify its advantages as follows;
•Theoretical contributions: TSL allows topics to act as unsupervised labels that represent global and target-specific word distributions as domain knowledge, and adapts PLMs to

---

[1] https://www.lemurproject.org/clueweb09.php/

resolve the gap and perform domain shift over topics.

•Practical contributions: As TAT updates only the target-specific word distributions, and does not need to update the parameters of PLMs, it generates more target-specific texts at lower computational cost than possible when using previous PLMs alone, while preserving PLM's advantages.

## 2 PREVIOUS WORK

Recently, pre-trained neural language models (NLMs), such as BERT (Devlin et al., 2019), GPT2 (Radford et al., 2019), XLNet (Yang et al., 2019), RoBERTa (Liu et al., 2019), and ALBERT (Lan et al., 2020) use Transformer (Vaswani et al., 2017) for learning contextualized text representations, and have yielded great advances in NLP tasks. Though achieving appealing performance, these Transformer-based models are better at exploring the relationships among local tokens than global semantics (e.g., word collocation over a given corpus) (Wang et al., 2020). As no Transformer-based model considers these explicit semantics, Wang et al. (Wang et al., 2020) rearranged and explored the semantics of topic models and developed a topic-friendly assistant for Transformer-based abstractive summarization models. UNIfied pre-trained Language Model (Dong et al., 2019) supports NLU and natural language generation (NLG) tasks by employing a shared Transformer network and utilizing specific self-attention masks to control which context the prediction is conditioned on. While He et al. improve (He et al., 2018) existing NMT models through layer-wise coordination of the encoder and decoder, and use modified attention masks in train both the encoder and the decoder simultaneously, their framework cannot be applied directly to domain-specific text generation; our framework differs from pre-learning in terms of task objectives, topic introduction, and fine-tuning. BertSUM (Wang et al., 2020) notes that topic models are better at learning explicit document semantics than Transformer. Different from their work, TAT aims to adapt NLMs to text generation tasks by performing domain shift.

As existing PLMs used large raw text data that do not necessarily contain sufficient knowledge or patterns that are directly related to the target-specific task, they still suffer from several potential limitations. More precisely, texts of specific task, e.g., movie review, can differ from PLMs training data (Chen et al., 2022). To address the question of whether pre-training on a corpus more directly tied to the task can further improve performance, continual pretraining (Gururangan & et al, 2020) has shown benefit of optimizing a PLM to a target domain before further finetuning. UDALM (Karouzos et al., 2021) first trains PLMs by masked language modeling (MLM) on the target domain and then trains a target classifier with source domain labeled data, while keeping the MLM objective on unlabeled target domain data. AdaPrompt (Chen et al., 2022) is a framework that can adapt a PLM for the end task considering both the prompts and the verbalizer, and adaptively continual pretraining on the retrieved data, which can benefit prompt-based methods on NLP downstream tasks.

While these training based approaches show the improvement in solving the domain gap, these practice of adapting and controlling pre-trained generative models poses the catastrophic forgetting: most approaches to enforcing a control objective result in a dramatic loss of capabilities of the original model beyond the scope of the control objective. As a way to take advantage of this achievement, we noted that some knowledge is universal across domains and some is not. Therefore, our approach aims to avoid this problem by incorporating a mechanism to recognize these relative differences, and intensively update only the target-specific knowledge.

As with global semantic information, topic models (Blei et al., 2003; Kawamae, 2018; Wang et al., 2020), and their extensions, take a global statistical view and look at the word distributions of topics across a given corpus; they represent each document as a bag-of-word (BOW) vector. Although these models organize a given corpus into small sets of prominent topics and have been proven to be powerful tools for uncovering latent structure, they and their application (Chang et al., 2021; Wang et al., 2018; 2020)are not, in the strict sense, sequence models.

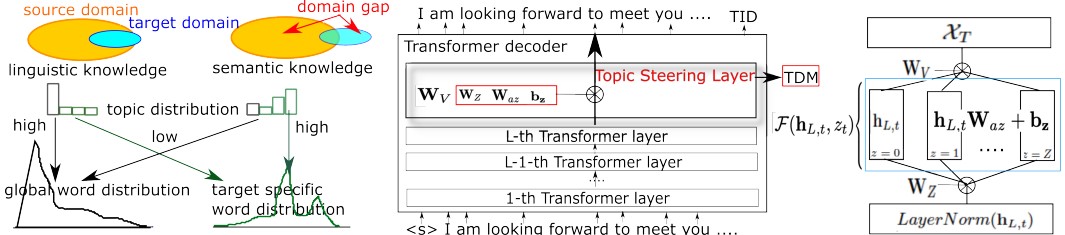

Figure 1: (left) The gap between the source and the target domain, domain gap, topic distribution, and word distributions, (center) The architecture overview of TAT that can adopt both NLMs and PLMs (e.g., GPT2), where their parameters are duplicated to form the TAT architecture. Newly introduced elements, Topic steering layer (TSL) and TDM, are highlighted in red, and TID denotes the representation of each text. (right) the detail of TSL with the affine in Eq (5). TAT learns $\mathbf{W}_V$, $\mathbf{W}_Z$, $\mathbf{W}_{az}$, and $\mathbf{b}_z$ and trains itself using TID and TDM on each text.

That is, we focus on bridging the domain gap as the domain shift, and aim to distill target-specific knowledge as topics, and steers PLMs toward these topics. This approach is designed to be compatible with training strategies while alleviating catastrophic forgetting.

## 3 METHODOLOGY

### 3.1 PROBLEM FORMULATION

Neural language models (NLMs) are trained as conditional language models for those specific tasks that require text generation (Bengio et al., 2003), PLMs. Given text sequence $\mathbf{x}_d = \{x_{d,1}, \cdots, x_{d,|x_d|}\}$ and dataset $D = \{\mathbf{x}_1, \cdots, \mathbf{x}_D\}$, NLMs are pre-trained by maximizing the following likelihood under forward autoregressive factorization:

$$\mathcal{L}_{LM}(\theta) = \sum_{d=1}^{|D|} \log P_\theta(x_d) = \sum_{d=1}^{|D|} \sum_{t=1}^{|x_d|} \log P_\theta(x_{d,t}|\mathbf{x}_{d,1:t-1}), \tag{1}$$

where $\theta$ represents model parameters.

As PLMs are trained on heterogeneous corpora, we can observe that the source and the target domain have both common (e.g., linguistic) and different (e.g., semantic) knowledge, as shown in Figure 1. Since the gap is intuitively in the difference between distributions over words (i.e., global and target-specific word distributions), and their distribution (i.e., topic distribution) over domains, we must recognize these distributions and train PLMs to update them. For example, given "My favorite artist is", PLM might predict "Michelangelo" as the next word, whereas the fine-tuned PLM yields "Botticelli". This leads us to introduce the latent variable of topic, $z$, into the NLMs and then modify Eq (1) to:

$$\mathcal{L}_{TLM}(\theta) = \sum_{d=1}^{|D|} \sum_{t=1}^{|x_d|} \log \sum_{z_t=1}^{Z} \underbrace{P_\theta(x_{d,t}|z_t, \mathbf{x}_{d,1:t-1})}_{\text{word distribution}} \underbrace{P_\theta(z_t|\mathbf{x}_{d,1:t-1})}_{\text{topic distribution}}, \tag{2}$$

where $z_t$ that indicates which global/target distribution is used, and $Z$ is the number of topics. Different from previous NLMs, topic language model (TLM) explicitly introduces topics into the generative process to utilize richer contextual information for improving NLMs/PLMs performance. Note that $P_\theta(z_t|\mathbf{x}_{d,1:t-1})$ is a multinomial distribution over discrete variables, not the Gaussian distribution used in variational autoencoders (Kingma & Welling, 2014) and its extensions such as (Wang & Wan, 2019; Zhu et al., 2021; Cai & Cai, 2022). This paper explores how to distill the domain-specific knowledge as topics, and steer PLMs toward topics.

### 3.2 MOTIVATION AND ARCHITECTURE DESIGN

As our challenge is to resolve the gap between the source and the target domain for the domain shift, it adapts PLMs to generate unconditional texts that reflect the target-domain

more than the source. The motivation of TAT is that domain adaptation should detect topics and update related distributions in the fine-tuning stage while preserving the semantic meaning and language structural information that pre-trained NLMs have, as discussed in 3.1. Following this motivation, $P_\theta(z_t|\mathbf{x}_{d,1:t-1})$ in Eq (2) can be interpreted as "the distribution over topics". $P_\theta(x_{d,t}|z_t, \mathbf{x}_{d,1:t-1})$ in Eq (2) is "the distribution over next words". They may be global or specific between the source and the target domain. The global is common over both domains and can be a distribution based on linguistic knowledge, while the specific is a distribution based on domain-specific knowledge. Although these differences are relative, vary with a given corpus, and cannot be clearly defined, the ratio of global word distribution in linguistic knowledge is intuitively considered much higher than that in semantic knowledge as shown in Figure 1; topics are designed to identify these different distributions and their weights.

These topics literally inherit the characteristics of latent topic models. While Transformer encodes context as local information, it requires large corpora to learn the higher-order and non-linear interactions between words, which demands more parameters, computation resources, and time. It is often observed that the learned attentive patterns of many heads are not as reasonable as we expect (Michel et al., 2019), and we might obtain this global information from the upper blocks by increasing the number of transformer blocks (Dosovitskiy et al., 2021); unfortunately, as the transformer architecture requires a large number of parameters, its computational cost is very high. Ramasesh et al. (Ramasesh et al., 2021) pointed that catastrophic forgetting occurs mainly in the higher layers. These insights lead us to place a topic steering layer (TSL) on the top Transformer layer and update only its related parameters to avoid this forgetting. This does not break any PLM structure, and allows reuse of PLMs and their parameters. As 1) a topic describes a co-occurrence pattern of tokens with similar semantics, and 2) the differences between pre-training and fine-tuning data sets exist not only in the topic itself, but also in the ratio of topics, we develop a training task, Topic Distribution Modeling (TDM), to align topics with each text.

Since global distributions do not require additional learning, our architecture is designed to find target-specific distributions through topics, and update them, $P_\theta(z_t|\mathbf{x}_{d,1:t-1})$ and $P_\theta(x_{d,t}|z_t, \mathbf{x}_{d,1:t-1})$, in fine-tuning. This design enables PLMs to emphasize knowledge that might otherwise have been buried, and so prevent catastrophic forgetting.

## 3.3 INPUT

Given a target domain corpus, TAT feeds the text as input to the decoder, as shown in Fig 1. Its layers convert the inputs into token (linguistic) embedding, and add special tokens [CLS], [SEP], [EOS], and . Following the text preprocessing of other Transformer-based NLMs, TAT tokenizes each input text to create the linguistic input of token embedding, where each sub-word is embedded with Word Piece (Wu et al., 2016) or another model-specific tokenizer (e.g., Byte-Pair Encoding (BPE) vocabulary (Radford et al., 2019)) whose length equals input length. [CLS] token is only inserted prior to the token, and denotes the class of each source text. [SEP] token is assigned to the end of each sentence in each input sequence, and indicates a sentence break. [EOS] token is assigned only after the last token in each input sequence.  token is only inserted prior to the token in each target text. Following other models, a learnable sequence position embedding is added to every input element indicating its order in the input sequence.

## 3.4 ATTENTION MECHANISM

Inside each Transformer layer, the previous layer's output $\mathbf{H}_{l-1} \in \mathbb{R}^{|x| \times d_h}$ is aggregated using multi-head self-attention, where $|x|$ is input sequence length. Thus the block core is multi-head attention with heads that use a causal mask to preclude attending to future tokens via the scaled dot-product attention:

$$\mathbf{Q} = \mathbf{H}_{l-1}\mathbf{W}_l^Q, \mathbf{K} = \mathbf{H}_{l-1}\mathbf{W}_l^K, \mathbf{V} = \mathbf{H}_{l-1}\mathbf{W}_l^V,$$

$$Attention(\mathbf{Q}, \mathbf{K}, \mathbf{V}) = softmax(\frac{\mathbf{Q}\mathbf{K}^T}{\sqrt{d}} + \mathbf{M})\mathbf{V}, \mathbf{M}_{ij} = \begin{cases} -\infty & \text{if } i < j, \\ 0 & \text{else} \end{cases}, \qquad (3)$$

where $\mathbf{W}_l^Q, \mathbf{W}_l^K, \mathbf{W}_l^V \in \mathbb{R}^{d_h \times d_k}$ are learnable weights for computing the queries, keys, and values, $\mathbf{Q}, \mathbf{K}, \mathbf{V} \in \mathbb{R}^{|x| \times d_k}$, respectively, $d_k$ is the shared dimensionality of the queries and keys. The self-attention mask, $\mathbf{M} \in \mathbb{R}^{|x| \times |x|}$, determines whether a position can attend to other positions, where, $M_{ij} = 0$ allows the $i$-th position to attend to the $j$-th position and $M_{ij} = -\infty$ prevents attending.

### 3.5 Topic steering layer (TSL)

As shown in Figure (1), TAT places these topics and their layer, Topic steering layer (TSL), on the top of Transformer layers to steer the text decoder. Then, TSL maps hidden representation vector $\mathbf{H}_L = [h_{L,1}, \cdots, h_{L,|x|}] \in \mathbb{R}^{|x| \times d_h}$ into topic vector $\mathbf{z} \in \mathbb{R}^Z$, and selects the topic-specific distribution over words on a given topic. This transformation yields Eq (2) by defining topic matrix, $\mathbf{W}_Z \in \mathbb{R}^{d_h \times Z}$, and word generation function, $\mathcal{F}(\mathbf{h}_{L,t})$, where $V$ is the size of the vocabulary. We apply these matrices to $\mathbf{h}_{L,t} \in \mathbb{R}^{d_h}$ in the text decoder, gain $\mathcal{X}_T$ and use it to sample the next token, $x_i$, according to the probability:

$$\mathcal{X}_T = LayerNorm(\mathbf{h}_{L,t})\mathbf{W}_Z \times \mathcal{F}(\mathbf{h}_{L,t}, z_t), \quad P_\theta(z_t|\mathbf{x}_{d,1:t-1}) \propto LayerNorm(\mathbf{h}_{L,t})\mathbf{W}_Z,$$

$$P_\theta(x_{d,t}|\mathbf{x}_{d,1:t-1}, z_t) \propto \mathcal{F}(\mathbf{h}_{L,t}, z_t), \quad P_\theta(x_{d,t}|\mathbf{x}_{d,1:t-1}) = \sum_{z_t=0}^{Z} P_\theta(x_{d,t}|\mathbf{x}_{d,1:t-1}, z_t)P_\theta(z_t|\mathbf{x}_{d,1:t-1}),$$

$$P(x_i \in \mathcal{X}_T) = \frac{exp(P_\theta(x_{d,t} = x_i|\mathbf{x}_{d,1:t-1}, z_t)/Te)}{\sum_i exp(P_\theta(x_{d,t} = x_i|\mathbf{x}_{d,1:t-1}, z_t/Te)}, \quad x_i \sim p(x_i \in \mathcal{X}_T)$$

(4)

where $\mathbf{W}_Z$ are learnable weights, $Te > 0$ is temperature and $x_i$ is the score of the $i$-th word in the vocabulary. As $\mathcal{X}_T$ is normalized into $p(x_i \in \mathcal{X}_T)$ to yield the probability over words, the next token is chosen by sampling a multinomial distribution with probabilities clipped to the top-$k$ tokens. Temperature-controlled stochastic sampling methods are used for generating text from trained NLMs or PLMs. While $Te \to 0$ approximates a greedy distribution, which magnifies the peaks in the probability distribution, $Te \to \infty$ flattens the distribution and makes it more uniform.

As with $\mathcal{F}(\mathbf{h}_{L,t}, z_t)$, we propose three transformations (addition, multiplication, and affine) to generate $x_{d,t}$ that accords with the given $z_t$ and $\mathbf{x}_{d,1:t-1}$, $\mathbf{h}_{L,t}$.

$$TID_i \propto \Sigma_t \mathbf{h}_{L,t} \quad \mathcal{F}(\mathbf{h}_{L,t}, z_t) = \mathbf{W}_V \times \begin{cases} \mathbf{h}_{L,t} & \text{residual if } z_t = 0 \\ (1-\omega)\mathbf{h}_{L,t} + \omega\mathbf{g}_z & \text{addition if } z_t = z \text{ and } z > 0 \\ \mathbf{h}_{L,t} \otimes \mathbf{g}_z & \text{multiplication if } z_t = z \text{ and } z > 0 \\ \mathbf{h}_{L,t}\mathbf{W}_{az} + \mathbf{b_z} & \text{affine if } z_t = z \text{ and } z > 0, \end{cases}$$

(5)

where $\mathbf{W}_V \in \mathbb{R}^{d_h \times V}$, $\mathbf{g}_z \in \mathbb{R}^{d_h}$, $\mathbf{W}_{az} \in \mathbb{R}^{d_h \times d_h}$, and $\mathbf{b}_z \in \mathbb{R}^{d_h}$ are the topic $z$ specific learnable weights. We prepare the residual to select the input if $z = 0$, take $\mathbf{h}_{L,t}$ as the global word distribution shown in Figure 1, which preserves PLM functionality, propose an alternative (i.e., addition, multiplication, and affine) for $z > 0$, and confirm by a pre-ablation analysis that affine is the best. Since both the ratio of the global words distribution and type of target-specific word distribution depend on the given target corpus, they are determined relative via fine-tuning using Eq (4). $TID_i \in \mathbb{R}^{d_h}$ denotes the average of $\mathbf{h}_{L,t}$ over each input, $i$-th text, without using $\mathbf{W}_z$.

**H1**

Note that just as Eq (1) is transformed into Eq (2) through the introduction of topics, the top layer of previous Transformer based NLMs/PLMs is decomposed into the product of $\mathbf{W}_Z$ and $\mathcal{F}(\mathbf{h}_{L,t})$ in Eq (4). While both $\mathbf{W}_Z$ and $\mathbf{W}_{az}$ ($\mathbf{b}_z$,$\mathbf{g}_z$) are newly introduced parameters, $\mathbf{W}_V$ is the existing parameter and updated in fine-tuning. Different from other Transformer-based PLMs, TAT 1) aligns the $t + 1$-th topic of target text, $P_\theta(z_t|\mathbf{x}_{d,1:t-1})$, and weights $P_\theta(x_{d,t}|\mathbf{x}_{d,1:t-1}, z_t)$ according to the distribution over topics, and 2) samples each token according to $p(x_i \in \mathcal{X}_T)$.

The top hidden state, $\mathbf{H}_L$, reflects the contextualized representation of the whole sequence in the decoder. As TAT applies the concept of topic to distill the target-specific knowledge as topics, the average of the token-level hidden states over each $i$-th text corresponds to a topic distribution of topic models, and N-gram topics by incorporating both the preceding topics and the topic specific distributions over words.

## 4 Model Training

Our training uses our newly introduced Topic Distribution Modeling (TDM)

### 4.1 Topic Distribution Modeling (TDM)

The objective of TDM is to minimize the difference between a document-specific topic distribution and a text specific representation, $TID$. The topic vector is the average of $z_t$ over each $d$-th text with Eq (4), $\mathbf{z}_d$, and is gained by multiplying the transpose matrix of $\mathbf{W}_Z$ used in Eq (4) to match the number of dimensions $\mathbf{z}_d$ and $TID_d$ on $d_h$. While we can make each a text-specific topic distribution, by computing the mean of $P_\theta(z_t|\mathbf{x}_{d,1:t-1})$ or max-over-time of all output topics, $\mathbf{z}_{1:t}$, we examine the best, the mean of distribution, in our comparison experiment. Since $\mathbf{z}_d$ learns representation directly through topics as shown in Figure 1, texts with similar content, as discussed in topic models, are considered to have similar topic distributions. As shown in Figure (1), TID is the final output of the last token of each input text sequence, and denotes the representation of each text.

For the metrics of L2 regression, cross-entropy, KL-Divergence, and triplet objective, our experiments confirm that TDM with triplet objective, $\mathcal{L}_{TDM}(\theta)$, attains better performance than the alternative combinations, and so employ it in our framework. Since there is no label that is the ground truth in unconditional text generation, we use $\mathbf{z}_d$ as the learning objective. Given $\mathbf{z}_d$ as an anchor representation of the $i$-th text, its corresponding text representation, $TID_d$, is taken as a positive embedding, while the other text representation, $TID_{\hat{d}}$ is taken as a negative embedding. Triplet loss tunes the model such that the distance between $\mathbf{z}_d$ and $TID_d$ is smaller than the distance between $\mathbf{z}_d$ and $TID_{\hat{d}}$. Mathematically, this objective minimizes the following loss function:

$$\mathcal{L}_{TDM}(\theta) = \max_{(\mathbf{z}_d, TID_d, TID_{\hat{d}}) \sim \mathbf{B}}(||\mathbf{z}_d - TID_d|| - ||\mathbf{z}_d - TID_{\hat{d}}|| + \epsilon, 0), \tag{6}$$

where $\mathbf{B}$ is each batch, $|| \bullet ||$ is a distance metric, and $\epsilon$ is the margin that ensures that $\mathbf{z}_d$ is at least $\epsilon$ closer to $TID_d$ than $TID_{\hat{d}}$; the sampling targets are batch units.

### 4.2 Training objective of TAT

We employ a unified multi-task learning framework that updates the decoder. As TAT can adapt PLM-based NLMs, their parameters, $\theta$, of Eq (2) are used to initialize the TAT decoder, and a fine-tuning process is employed to adapt $\theta$ to the fine-tuning data. To optimize these parameters and bridge the gap between the data used in the pre-training and the fine-tuning process, we optimize the model loss in this tuning process. Using Eq (2),(6), we can define the loss function, $\mathcal{L}(\theta)$, as the sum of these objective functions that is to be optimized in the fine-tuning stage:

$$\mathcal{L}_{TAT}(\theta) = -\mathcal{L}_{TLM}(\theta) + \lambda_{TDM}\mathcal{L}_{TDM}(\theta), \tag{7}$$

where $\theta$ is the parameter set of TAT, $\lambda_{TDM}$ are hyper parameters to balance the importance of TLM and TDM. We use Adaptive Moment Estimation (Adam) (Kingma & Ba, 2015) over mini-batches to update parameters, and adopt the dropout strategy (Srivastava et al., 2014) to optimize networks.

## 5 EXPERIMENTS

### 5.1 Datasets and Experiment design

**Datasets** We conducted evaluations using Amazon review[2] and Yelp[3], as they are large publicly available datasets and are manually evaluable for screened colleagues. We used these datasets because the resulting data size is computationally feasible on a general-purpose server, includes a variety of topics that are different from pre-training corpus, meets

---

[2]`https://huggingface.co/datasets/amazon_reviews_multi`
[3]`https://www.yelp.com/dataset/download`

Table 1: Basic statistics of the datasets used in this paper: In the attributes, $\#Z$ denotes the number of topics for evaluating topic coherence.

| Dataset | #reviews | #vocabulary | #Z |
|---|---|---|---|
| **Amazon** | 210,000 | 246,534 | 100 |
| **Yelp** | 6,685,900 | 365,762 | 200 |

the public reproducibility requirement, and can validate our insight that a small corpus can provide significant benefits (Gururangan & et al, 2020). More precisely their domain similarity is far from PLMs and high enough that they can be used as additional training data for each other in domain-adaptive pretraining (DAPT) (Gururangan & et al, 2020). Each record in the dataset contains a review text, review title, star rating, anonymized ID, and coarse-grained product category, we use only review texts. All reviews were truncated after 2,000 characters, and all reviews were at least 50 characters long. Among the languages present, we used only English for ease of interpreting the results. We applied the same pre-treatment to the Yelp data set and statistics of the resulting data set are shown in Table 1. We used 90%, 5%, and 5% of each data set as training (e.g., text generation), validation and test sets, respectively. The final performance comparison results are derived from the test set, which corresponds to task-adaptive pretraining (TAPT) (Gururangan & et al, 2020).

**Experiment Setup** We implemented TAT by using Pytorch 1.7.1[4] and will release this code soon. We set $\epsilon$ in Eq (6) to 0.2, and $\lambda_{TDM}$ in Eq (7) to 0.5. As the average length of each text used in fine-tuning data set is around 60, we set the maximum input sequence length to 64. Note that the ground truth texts were excluded from training/validation data to prevent information leakage. TAT uses GPT2 as the PLM. Following the training setting, we used Adam with $\beta_1 = 0.9$, and $\beta_2 = 0.999$ was used for optimization, over mini-batches to update parameters; the dropout strategy (Srivastava et al., 2014) was adopted for network optimization. The learning rate was 3e-5, with linear warmup over the first 500 steps and linear decay, where we set the dropout rate, the weight decay, and the batch size to 0.1, 0.01, and 256, respectively. We fine-trained all models on 8 Nvidia Tesla V100 GPUs, each with 32G memory.

## 5.2 Topic Coherence

This experiment aims to evaluate how well TAT discovers topics, and compare TAT with existing topic-based embedding models. In order to quantitatively assess topic quality, we use the topic coherence measure (Mimno et al., 2010) to examine the relatedness of the top-ranked words. This score shows high consistency with human judgements in terms of topic quality (Mimno et al., 2010), where higher scores indicate greater topic coherency. As the baselines, we used NMF (Lee & Seung, 1999), and LDA+TWE (TWE-1)[5], Topic2Vec[6], with CLM[7]. We set the iterations of the Gibbs sampler, parameter update or epochs to 200 for all models except TAT, where the first 50 iterations were used to burn in the Gibbs sampler; CLM used matrix factorization in learning word embedding representation. We varied the number of top ranked words, and measured the resulting performance by using the coherence model function of gensim[8] with "u_mass".

Table 2(left) shows that the top words of the learned topics are semantically coherent, which coincides with the finding that using word embedding improves the quality of topic models (Liu et al., 2015; Xun et al., 2017; Nguyen et al., 2015). TAT learns topics using the order of both word and topic in each document and updates them iteratively, groups semantically-related words more efficiently than the alternative approaches, and yields more distinct topics.

---

[4]https://pytorch.org/

[4]https://huggingface.co/transformers/pretrained_models.html

[5]https://github.com/largelymfs/topical_word_embeddings

[6]https://github.com/ukgovdatascience/topic2vec

[7]https://github.com/XunGuangxu/2in1

[8]https://radimrehurek.com/gensim/models/coherencemodel.html

Table 2: (left)Comparison of topic coherence, (right)Example of topics discovered by TAT: $N$ is #ranked words. In this table, T2V denotes Topic2Vec. The size of the embedding space for Topic2Vec, CLM and TAT was set to 100(Amazon)/200(Yelp), the skip length and #negative sampling were set to 5 and 5, respectively. The values in bold show best performance, and denotes the statistical significance for $p < 0.01$, compared to the best baseline. These words were selected from their distance from the topic embedding.

| N | **Amazon** 5,10,20 | **Yelp** 5,10,20 | | topic 1 | topic 2 | topic 3 | topic 4 |
|---|---|---|---|---|---|---|---|
| TWE | -2.11,-2.26,-3.33 | -2.93,-3.08,-3.76 | | iphone | small | action | excellent |
| NMF | -1.92,-2.54,-3.23 | -2.67,-2.91,-3.82 | | os | clear | sci-fi | dark |
| T2V | -1.75,-2.13,-2.61 | -2.28,-2.36,-2.71 | | touch | better | visual | cool |
| CLM | -1.71,-1.82,-2.26 | -1.52,-1.82,-2.22 | | camera | long | staring | pleasing |
| TAT | **-0.88**,**-1.23**,**-1.77** | **-0.92**,**-1.45**,**-1.01** | | battery | fantastic | effect | classic |

Table 3: Comparison and Ablation analysis of various adaptation methods: In this table, F, P, T, D-N, B-N, M and RL denote Fluency, Perplexity, Topic, Dist-N, BLEU-N, METEOR, and Rouge-L respectively. In each row, upper/lower value is Amazon/Yelp. GPT-2+f, +pt, and +DT is GPT-2 after fine-tuning, with prefix-tuning (Li & Liang, 2021), and with DAPT+TAPT (Gururangan & et al, 2020), respectively. TAT freezes the parameters of GPT-2 to evaluate the effect of TSL and TDM, TAT+DT freezes them after DT. The bold values have equivalent meaning to its usage in Table 2.

| Model GPT-2 | | | F ↑ | P ↓ | T ↑ | D-2 ↑ | D-3 ↑ | B-2 ↑ | B-3 ↑ | M ↑ | RL ↑ |
|---|---|---|---|---|---|---|---|---|---|---|---|
| +f | | | 3.11 | 27.32 | 0.73 | 5.12 | 10.02 | 14.48 | 7.82 | 10.23 | 17.36 |
| | | | 2.65 | 32.32 | 0.68 | 4.55 | 9.03 | 14.21 | 7.22 | 9.67 | 16.11 |
| +pt | | | 3.08 | 27.38 | 0.73 | 5.02 | 10.02 | 14.93 | 7.88 | 10.83 | 17.8 |
| | | | 2.63 | 32.34 | 0.68 | 4.57 | 9.06 | 14.33 | 7.25 | 10.45 | 17.21 |
| +DT | | | 3.26 | 26.12 | 0.75 | 5.23 | 10.31 | 15.63 | 8.12 | 11.23 | 18.78 |
| | | | 2.63 | 32.34 | 0.68 | 4.57 | 9.06 | 14.33 | 7.25 | 10.95 | 17.78 |
| TAT | \|Z\| | TDM | | | | | | | | | |
| | 10 | w | 3.45 | 25.16 | 0.79 | 5.64 | 11.18 | 17.61 | 8.88 | 12.21 | 21.14 |
| | 10 | w | 2.85 | 29.88 | 0.75 | 4.67 | 9.52 | 15.38 | 7.71 | 11.78 | 20.05 |
| | 20 | w | **3.56** | **23.21** | **0.81** | **5.91** | **12.12** | **18.13** | **9.12** | **13.14** | **22.15** |
| | 20 | w | **2.97** | **27.23** | **0.78** | **5.06** | **10.25** | **16.42** | **8.38** | **12.16** | **20.98** |
| | 20 | w/o | 3.41 | 25.21 | 0.78 | 5.62 | 11.12 | 17.58 | 8.81 | 12.12 | 20.85 |
| | 20 | w/o | 2.81 | 29.91 | 0.74 | 4.62 | 9.47 | 15.35 | 7.69 | 11.72 | 19.92 |
| +DT | 20 | w | 3.45 | 25.16 | 0.79 | 5.64 | 11.18 | 17.61 | 8.88 | 12.21 | 21.14 |
| | 20 | w | 2.85 | 29.88 | 0.75 | 4.67 | 9.32 | 15.38 | 7.71 | 11.78 | 20.05 |

We show examples of discovered topics in Table 2, where the ID of the topic is arbitrary; topic2/4 co-occurs with topic1/3. This result shows that topic 1 and topic 3 have many words associated with products and contents, while topic 2 and topic 4 have many subjective words corresponding to features. This indicates that TAT extracts interpretable topics.

## 5.3 TEXT GENERATION

**Baselines:** To evaluate the quality of generated texts, we add the latest adaptation approach, GPT2 with prefix-tuning (Li & Liang, 2021), DAPT+TAPT (Gururangan & et al, 2020).

**Automated evaluation:** We used test-set perplexity, Dist (Li et al., 2016), and BLEU-N (Papineni et al., 2002) metrics to measure performance, METEOR (Lavie & Agarwal, 2007), and ROUGE (Lin, 2004) metrics to measure performance (Sai et al., 2023). Perplexity is an automated measure of fluency, and while its effectiveness has been questioned in open-domain text generation (Liu et al., 2016), we use the well-known test-set perplexity using different pre-trained NLMs. $n$-gram based metrics (Dist, BLEU, METEOR, ROUGE) count the overlap between the generated text, and its corresponding reference text in the test data after removing seed words.

Table 4: Case study for Amazon: (top) Ground Truth, (center) GPT2+TAPT+DAPT, and (TAT). We set seed words of "I am disappointed in this purchase", and show the text generated by each model.

| I am disappointed in this purchase. I bought one of these in another color and in size XL |
| --- |
| The color is not as vibrant as I would like. It does however still look great. I will use |
| I ordered an XL size in black which arrived with a large hole. There's no way anyone |

**Human evaluation:** We employ fluency testing on attribute relevance as the human annotation (Dathathri et al., 2020). Annotators were asked to evaluate the fluency of each individual sample on a scale of 1-5, with 1 being, not fluent at all, and 5 being, very fluent, as done in (Lample et al., 2019). Topic reports the fraction of samples matching the target domain as evaluated by the manual annotators. To consistently evaluate generated texts, we recruited and screened 10 colleagues who were familiar with movies, music (Amazon), and restaurants (Yelp) and who could interpret reviews. H3

**Comparisons:** As shown in Table 3, TAT outperformed the baselines and achieved better performance over both data sets. These results support our hypothesis that TSL allows TAT to distill knowledge in the form of topics, and update only the target-specific word distributions to prevent catastrophic forgetting. The decline due to the introduction of DT into TAT implies catastrophic forgetting.

**Ablation analysis:** To investigate the respective contributions of TAT components, (i.e., TSL and TDM), we conducted an ablation analysis. We removed different components and the resulting text generation quality is shown in Table 3. This table shows that the complete setting of TAT achieves better performance across both datasets. A within table comparison shows that TSL is most effective. By comparing the effects of TSL and TDM for the same pre-training data, these newly introduced tasks improve text generation performance, while TL was more effective than TDM and DAPT+TAPT.

**Error analysis:** A manual error analysis showed that some instances marked as errors were in fact assessed correctly as allowed by partial matching of words in a text. When the ground truth text is personalized, human judgement is difficult even if the generated text is different from the ground truth, see Table 4. where the generated texts included more abstract or higher frequency words than the reference sentences.

## 6 DISCUSSION

As TAT adapts PLMs toward the target domain as domain shift, it focuses on the difference between the source and the target domain as word distributions, uses topics as a memory and an indicator to select these distributions through the use of TSL and TDM. Topics are based on word co-occurrence in the texts, which varies with the dataset, and allows TAT to capture the global information. Previous NLMs use local context information to capture semantic meaning. Formally, TAT performs domain shift by recalculating $P_\theta(x_{d,t}|z_t, \mathbf{x}_{d,1:t-1})$ and aligning the $P_\theta(z_t|\mathbf{x}_{d,1:t-1})$ with the target domain, see Eq (2) and Figure1.

TAT mitigates the gap between the source and the target by highlighting the target-specific word distributions through topics and updating only these distributions, even if $Z$ is small. The number of parameters in $\mathbf{W}_Z \in \mathbb{R}^{d_h \times Z}$, $Z \times \mathbf{W}_{az} \in \mathbb{R}^{d_h \times d_h}$, and $Z \times \mathbf{b}_z \in \mathbb{R}^{d_h}$, newly introduced by TSL, is a much smaller number of parameters than PLMs, and fixing PLMs's parameters in the fine-tuning stage lowers the overall computational cost; its effectiveness is shown in Table 3. Just as topic models have enjoyed success in areas other than text processing, fine-tuned PLMs with TAT could be applied to other tasks. H4

## 7 CONCLUSION

The proposal of this paper, TAT, can adapt PLMs to the unconditional text generation task while using the target domain as a constraint. Experiments showed that the components of TSL and TDM enable TAT to discover target-domain-specific topics, fine-tune PLMs over these topics, and generate valid texts reflecting a given small fine-tuning data set.

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

## A    APPENDIX

