# OpenReview forum: "Topic Aware Transformer: Domain Shift for Unconditional Text Generation Model"
_ICLR.cc/2023/Conference — Submitted to ICLR 2023_

### Official Review · Reviewer_qrSx · 2022-10-24

**Confidence:** 4
**Correctness:** 3
**Technical Novelty And Significance:** 3
**Empirical Novelty And Significance:** 3
**Recommendation:** 6

**Clarity, Quality, Novelty And Reproducibility:**

This paper proposes a novel idea, with detailed introduction and good evaluation.
The selection of datasets and the future code release make it easy to reproduce.
There are some points can be further clarified:
1. What is the catastrophic forgetting while adapting PLMs? The authors cite a lot of prior work without giving clear enough explanations.
2. How to define "global word distributions" and "target-specific word distributions" in Sec 3.1? As mentioned in sec 3.2, there might not be very clean definitions, but showing examples can be better.
3. Figure 1 left: what is "large" and "small" refer to?
4. "The topic vector is the average of zt over each i-th text with Eq (4)" in sec 4.1. What does "each i-th text" mean? Each token or each input text? Writing an equation can help with such ambiguous wording.
5. What is the motivation for the TDM loss? The authors say "align topics with each text", but what is the meaning of aligning the final predicted topics (TID) with previously predicted topics, since TID itself is not related to token generation.
6. What is the dynamic of topic prediction during inference? Are there any shifting of topics? More discussion on this can be interesting.


**Strength And Weaknesses:**

Strength:
1. This proposed idea is novel, interesting and well-motivated.
2. The resulted models can bring us more interpretability of the process of language modeling, by predicting topics before predicting next tokens.
3. The evaluation on Yelp and Amazon datasets are comprehensive.

Weakness:
1. It seems that the proposed model can and should go beyond unconditional generation tasks. After introducing an extra TSL layer and learning meaningful topics, it seems unconditional generation tasks cannot fully leverage the potential of learned models. If we treat the learned topic distributions as learned interpretable latent variables, it should be easy and interesting to inject more topic control into the generation process.
2. While "alleviating catastrophic forgetting" is one of the authors' motivations, discussion and comparison of tuning-free adaption is missing. For example, Plug and Play Language Models: A Simple Approach to Controlled Text Generation (https://arxiv.org/pdf/1912.02164.pdf), and FUDGE: Controlled Text Generation With Future Discriminators (https://aclanthology.org/2021.naacl-main.276.pdf). By such tuning-free controlled generation approach, we can also easily adapt PLM to target domains by changing the predicted word distributions.
3. The presentation can be improved, such as the form of citation makes section 2 really hard to follow. The left part and the right part of figure 1 are also hard to understand.

**Summary Of The Paper:**

This paper proposes a topic aware transformer (TAT) to adapt pretrained language models to target domains through topic modeling. The motivation is, to alleviate forgetting, TAT explicitly quantify the domain shift as topic shifts. By introducing a topic steering layer (TSL) on top of transformers, TAT decompose the task of predicting next tokens into two tasks: (i) Firstly, predict the distributions of topics given previous tokens. (ii) Secondly, shift the word distribution according to predicted topics and then predict the next token. Combining the TSL with a newly introduced topic distribution modeling (TDM) loss, which aims at aligning the topic prediction, the proposed TAT achieves better performance of unconditional generation on two datasets, comparing to DAPT+TAPT and prefix tuning.

**Summary Of The Review:**

I think this paper proposed an interesting and novel idea, with good experiments.
My biggest concern is the missing of comparisons to previous work like PPLM and FUDGE, which makes it harder to judge the real contribution of this work.
I also suggest there should be more tasks to play with the proposed model.

---

> ### Author Response · Authors · 2022-11-08
> **Answers and corrections for further discussion**
>
> Thank you for reviewing our manuscript.
> We submitted a manuscript based on the questions and comments we received, where we have highlighted the corrections in red and the references in red underline.
>
> W1) It seems that the proposed model can and should go beyond unconditional generation tasks. After introducing an extra TSL layer and learning meaningful topics, it seems unconditional generation tasks cannot fully leverage the potential of learned models.\
> --->We believe so. Since there is so much controversy in its application to the most rudimentary tasks, the results of its application to other tasks will be made public once this manuscript is accepted.
>
> W2) While "alleviating catastrophic forgetting" is one of the authors' motivations, discussion and comparison of tuning-free adaption is missing.\
> --->As stated in 6, H4, our approach also freezes the PLM's parameters and updates topic related parameters in the fine-tuning process.
> Formally, when $z$=0 in Eq (5), it corresponds to ``residual'' and preserves PLM functionality.
> While both PPLM and FUDGE are language models,
> they differ our framework in guiding PLMs towards desired attribute and belong to conditional language models.
> When our framework is extended to conditional text generation tasks,
> we will compare it with PPLM, FUDGE, Prefix-Tuning, NRP and COCON.
>
> Prefix-Tuning: Xiang Lisa Li and Percy Liang. 2021. Prefix-Tuning: Optimizing Continuous Prompts for Generation. In ACL/IJCNLP. 4582–4597.\
> NRP: Fredrik Carlsson, Joey Öhman, Fangyu Liu, Severine Verlinden, Joakim Nivre, and Magnus Sahlgren. 2022. Fine-Grained Controllable Text Generation Using Non-Residual Prompting. In ACL. 6837–6857.\
> COCON: Alvin Chan, Yew-Soon Ong, Bill Pung, Aston Zhang, and Jie Fu. 2021. CoCon: A Self-Supervised Approach for Controlled Text Generation. In ICLR.
>
> W3) The presentation can be improved, such as the form of citation makes section 2 really hard to follow. The left part and the right part of figure 1 are also hard to understand.\
> --->Apologies for leaving the citation as ~\cite instead of ~\citep. We have corrected it and create new figure in the current manuscript.
>
> Q1) What is the catastrophic forgetting while adapting PLMs? The authors cite a lot of prior work without giving clear enough explanations.\
> --->Existing learning method overwrites PLMs' parameters with updates or even partial updates to the original parameters, and lead to the loss of information that PLMs have learned from a large source domain.
> We call this loss, the catastrophic forgetting.
>
> Q2) How to define "global word distributions" and "target-specific word distributions" in Sec 3.1? As mentioned in sec 3.2, there might not be very clean definitions, but showing examples can be better.\
> --->Although word examples are included in the Table 2, we decided that other descriptions would be more effective in effectively and qualitatively articulating our hypothesis. We will adjust the paper and attempt to republish them in the next submitted version.
>
> Q3) Figure 1 left: what is "large" and "small" refer to?\
> --->We change "large" and "small" to ''high'' and ''low'' that is probability over "global word distribution" and "local word distributions" in the linguistic knowledge, and semantic knowledge, respectively.
>
> Q4) "The topic vector is the average of zt over each i-th text with Eq (4)" in sec 4.1. What does "each i-th text" mean? Each token or each input text? Writing an equation can help with such ambiguous wording.\
> --->Apologies for the ambiguous notation, $i$-th->$d$-th. This notation is input text. We have corrected it in the current manuscript.
>
> Q5) What is the motivation for the TDM loss? The authors say "align topics with each text", but what is the meaning of aligning the final predicted topics (TID) with previously predicted topics, since TID itself is not related to token generation.\
> --->Because our task is to generate unconditional text generation, and has no guidances that steer PLMs like the conditional text generation,
> the motivation of TDM loss is to align each text on the semantic level.
> As this semantic corresponds to the topic distribution,
> we are hinted by a contarstive learning and then design to apply it into measuring the semantic similarity between texts.
>
> Q6) What is the dynamic of topic prediction during inference? Are there any shifting of topics? More discussion on this can be interesting.\
> --->This dynamic and the size of topic shift depends on both the number of topics, $Z$, and the differences between the source and target domains. This dynamic can be observed as the value change of $W_{Z}$.
> If the corresponding topic exists in the source domain, TAT increases the probability of that topic through learning.
> If the corresponding topic does not exist in the source domain, TAT updates the topic distribution by learning a new topic and adding it to the existing topics.
> The hierarchy of topics or the automatic determination of their number is the next topic.

---

> > ### Comment · Reviewer_qrSx · 2022-11-17
> > **Thanks for the response**
> >
> > Thank you for your detailed response! Some of my concerns are clarified.
> > There is no significant change in my judgment of this paper. I keep my score.
> >
> > As pointed out by other reviewers, some parts of the current manuscript might be hard to follow. In general, I encourage more revision to make it a good paper.

---

> > > ### Author Response · Authors · 2022-11-18
> > > **Thanks for your response**
> > >
> > > We appreciate your response and will use your comments meaningfully in our revisions.

---

### Official Review · Reviewer_DzP1 · 2022-10-24

**Confidence:** 3
**Clarity, Quality, Novelty And Reproducibility:** 1. Some parts of the equation are unc…
**Correctness:** 3
**Technical Novelty And Significance:** 3
**Empirical Novelty And Significance:** 3
**Recommendation:** 6

**Strength And Weaknesses:**

Strength:
1. The paper proposes a new topic-aware transformer to resolve the domain shift for unconditional text generation. The motivation of the paper is clearly stated. The TAT achieves the best performance for topic coherence and text generation. The whole idea is interesting.

2. The paper conduct both automatic and human evaluation. The paper also did some ablation analysis to show the contribution of each component. The paper also presents some generation results as examples.


Weaknesses:
1. Some parts of the paper are not clearly written which will be discussed in the clarity section.

2. Some topic-related papers are not covered in the related work such as Zhu et al., (2021), and Chang et al., 2021.

3. The citation style is not correct. Authors need to use citep{} The paper uses large space to squeeze tables and captions, making readers hard to read. Authors should not abuse space.

4. The evaluation metrics used in the paper are a little bit old and only concentrate on the token overlaps. The paper needs to include some newer metrics such as BERTscore (Zhang et al., 2019), and BARTScore (Yuan et al., 2021) which can check semantic similarity.


Zhu, L., Pergola, G., Gui, L., Zhou, D., & He, Y. (2021). Topic-driven and knowledge-aware transformer for dialogue emotion detection. arXiv preprint arXiv:2106.01071.

Chang, H. S., Yuan, J., Iyyer, M., & McCallum, A. (2021). Changing the mind of transformers for topically-controllable language generation. arXiv preprint arXiv:2103.15335.

Zhang, T., Kishore, V., Wu, F., Weinberger, K. Q., & Artzi, Y. (2019). Bertscore: Evaluating text generation with bert. arXiv preprint arXiv:1904.09675.

Yuan, W., Neubig, G., & Liu, P. (2021). Bartscore: Evaluating generated text as text generation. Advances in Neural Information Processing Systems, 34, 27263-27277.

**Summary Of The Paper:**

This paper proposes a new topic-aware transformer that alleviates the gap between the pretrained language model and the target domain. The paper introduces an additional topic latent variable $z$ as an additional condition when generating new tokens. The new topic latent variables come from the topic steering layer, which maps hidden representations into the topic vector. The model is optimized with both topic distribution modeling and maximum likelihood. The backbone of the paper is GPT2. The model is tested on two datasets: Amazon and Yelp. The paper checks the topic coherence and its text-generation ability  The paper did both automatic and human evaluation. The paper also did additional ablation analysis and case study.

**Summary Of The Review:**

Overall, the proposed new model: the topic-aware transformer is quite interesting and follow s motivation. However, some parts of the paper are not very clear. The experiment section and paper format need to be improved.

---

> ### Author Response · Authors · 2022-11-08
> **Answers and corrections for further questions**
>
> Thank you for reviewing our manuscript.
> We submitted a manuscript based on the questions and comments we received, where we have highlighted the corrections in red and the references in red underline.
>
> W1) Some topic-related papers are not covered in the related work such as Zhu et al., (2021), and Chang et al., 2021.\
> --->We have quoted them in the current manuscript.
>
> C1) The evaluation metrics used in the paper are a little bit old and only concentrate on the token overlaps. The paper needs to include some newer metrics such as BERTscore (Zhang et al., 2019), and BARTScore (Yuan et al., 2021) which can check semantic similarity.\
> --->Since GPT-2, which we used, uses a different architecture than BERT/BART as well as a different tokenizer, we are examining how we can use or modify those metrics. If you have any literature that we should refer to, it would be very helpful.
>
> C2) in equation 5, where does z ome from? Although readers can later learn that z actually comes from
> , it would be better to split equation 4 into two formulas.\
> --->We have corrected it in the current manuscript.
>
> Q1) Authors also need to rewrite equation 5, because the definition of F
>  is unclear. Which transformation does the paper finally used in the model?\
> --->As stated in 3.5, H1 and shown in Figure 1, we use finally ``affine'' after ablation analysis over these transformations.
>
> C3) The human evaluation details are not clear. For example, the number of annotators in the human evaluation is unknown. The annotation guidelines should be attached to the appendix.\
> --->As stated in 5.3, H3, Human evaluation, we recruited and screened 10 colleagues who were familiar with movies, music (Amazon), and restaurants (Yelp) and who could interpret reviews.
> We shared these guidelines via audio and email, but are being edited for publication.
>
> C4) However, the paper doesn't provide any code or dataset for reproduction.\
> --->We will release the code after acceptance of this paper, but use publicly available datasets without special treatment.

---

> > ### Comment · Reviewer_DzP1 · 2022-11-17
> > **Thank you for the response**
> >
> > Thank you very much for addressing my comments.
> > The BERTScore and BARTScore metrics can take in plain text and works as regular BLEU, ROUGE, etc. Therefore, I don't think the tokenizer and model architecture will be a barrier.
> > Overall, I think the paper might still need further revision.

---

> > > ### Author Response · Authors · 2022-11-17
> > > **Thank you and confirmation**
> > >
> > > Thank you for your comment.
> > > As you know, BERTScore, and BARTScore use BERTS, and BART as its backbone, takes candidate (target) and reference as its input,
> > > and yields their tokens as its output.
> > > So you mean we can use generated texts via GPT-* as candidates for these metrics?

---

> > > > ### Comment · Reviewer_DzP1 · 2022-11-19
> > > > **Reply**
> > > >
> > > > Yes, so you can still run those experiments. Thank you!

---

### Official Review · Reviewer_cPS3 · 2022-10-25

**Confidence:** 3
**Correctness:** 2
**Technical Novelty And Significance:** 2
**Empirical Novelty And Significance:** 2
**Recommendation:** 3

**Clarity, Quality, Novelty And Reproducibility:**

Here are my questions on the method section:
1. In equation 2, it seems t represents the index of tokens, which aligns with the normal notation for token position. However, in equation 5, it seems that t represent the index of topics. Why do you use the same symbol for two concepts?
2. In equation 5, you have three kinds of F function for z>0, then which one do you use finally?
3. Why do you need to describe the attention mechanism in section 3.4 in such details?
4. In section 4.1, what do you mean by "while the other text representation, TID is taken as a negative embedding"?
5. In equation 5, TID is the average of h, which are the representation vectors after L layers of transformer blocks, while z is the latent variable of topics. Then why do we want to minimize the distance between these two vectors?

Here are my questions on the results section:
1. In Table 1, the number of Z is set to 100 and 200 for the two datasets. How do you decide it?
2. What are the details of those baselines used in section 5.2 and 5.3? Could you elaborate them instead of just giving references or links?
3. What is the topic coherence measure?
4. In table 3, whar is |z|?

There are too many abbreviations that look very similar in the paper, which makes it very hard for readers to understand and follow.

**Strength And Weaknesses:**

Strengths:
1. The general idea of modeling topics as latent variables for domain adaptive pre-training does make sense.
2. The results look good.

Weaknesses:
1. The format of this paper has been totally changed. I think it is not allowed to change the template. The font looks weird. And those table layouts look very condense and hard to read.
2. The notations of equations look very confusing. And the  descriptions of methods are also hard to follow. I will pose all my questions on the method section in the following section.
3. Since I cannot fully follow the method, I have concerns on the effectiveness of the method.
4. In the results section, those baselines have not been well described so it is impossible for me to judge whether the baselines are enough. Even the metric used for evaluating the topic coherence is not described in enough details so I do not know whether this metric is suitable for the task or not.

**Summary Of The Paper:**

This work proposes to further fine-tune an unconditional language model on a domain-specific corpus via modeling the topics as a discrete latent variable.

**Summary Of The Review:**

The method proposed may be good but the bad writing makes it hard for readers to discover its significance.

---

> ### Author Response · Authors · 2022-11-08
> **Answers and correction for deep understanding**
>
> Thank you for reviewing our manuscript. We submitted a manuscript based on the questions and comments we received, where we have highlighted the corrections in red and the references in red underline.
>
> W1)The format of this paper has been totally changed. I think it is not allowed to change the template. The font looks weird. And those table layouts look very condense and hard to read.\
> --->Apologies for leaving the citation as ~\cite instead of ~\citep.　We have not changed the format and style file, but it is possible that the fonts have been switched in our environment.
> Now, we submit the manuscript created using the overleaf.
> If necessary, we will submit a complete set of source files to prove it.
>
> Q1) In equation 2, it seems t represents the index of tokens, which aligns with the normal notation for token position.
> However, in equation 5, it seems that t represent the index of topics. Why do you use the same symbol for two concepts?\
> --->$t$ in $z_{t}$ represent the topic in $t$-th token rather than index of topics.
> As each token has its own topics and $z_{t}$ denotes the topic of $x_{t}$, we use the same symbol.
>
> Q2) In equation 5, you have three kinds of F function for z>0, then which one do you use finally?\
> --->As stated in 3.5, H1 and shown in Figure 1, we use finally ``affine'' after ablation analysis over these transformations.
>
> Q3) Why do you need to describe the attention mechanism in section 3.4 in such details?\
> --->Since it is directly related to the core of the idea, we describe it for readers to refer to.
>
> Q4) In section 4.1, what do you mean by "while the other text representation, TID is taken as a negative embedding"?\
> --->As both $z_{d}$ and $TID_{d}$ are separate expressions of $d$-th text,
> $TID_{\acute{d}}$ is the expressions of $\acute{d}$-th text, different text.
>
> Q5) In equation 5, TID is the average of h, which are the representation vectors after L layers of transformer blocks, while z is the latent variable of topics. Then why do we want to minimize the distance between these two vectors?\
> --->As stated in 4.1, H2, we multiply $z$ by the transpose matrix of W_{Z} used in Eq (4) to match the number of dimensions zi and TIDi on d_{h}.
>
> Q6) In Table 1, the number of Z is set to 100 and 200 for the two datasets. How do you decide it?\
> --->We changed the number of topics in 10 intervals from 50 to 250, and compare them on each number.
> This table shows the results for other models to yield relatively good performance,
> where these results corresponding to 100, and 200, respectively.
>
> Q7) What are the details of those baselines used in section 5.2 and 5.3? Could you elaborate them instead of just giving references or links?\
> --->Of course it is possible, but it would be helpful if you could give us specific instructions to answer your request, as it seems to contradict your question, Q3.
>
> Q8) What is the topic coherence measure?\
> --->Omitted due to space limitations, but used the following equation defined in the cited, Mimno et al. (2010)
> Letting $D(v)$ be the document frequency of word type $v$ (i.e., the number of documents with least one token of type v) and $D(v, v′)$ be co-document frequency of word types $v$ and $v′$ (i.e., the number of documents containing one or more tokens of type $v$ and at least one token of type $v′$), we define topic coherence as
> $C(t;V^{(t)})=\sum_{m=2}^{M}\sum_{l=m-1}^{M}log \frac{D(v_{m}^{(t)},v_{l}^{(t)})}{D(v_{l}^{(t)})}$,
> where $V(t)=(v(t),...,v(t))$ is a list of the $M$ most probable words in topic t.
> A smoothing count of 1 is included to avoid taking the logarithm of zero.
>
> Q9) In table 3, whar is |z|?\
> --->The number of topics.

---

### Official Review · Reviewer_6wEU · 2022-10-26

**Confidence:** 3
**Correctness:** 3
**Technical Novelty And Significance:** 2
**Empirical Novelty And Significance:** 2
**Recommendation:** 3

**Clarity, Quality, Novelty And Reproducibility:**


The clarity of the paper is low. The mathematical notation seems to be used in an inconsistent way, leading to impossible equations. There are also many grammatical errors, making understanding even more difficult.
For example, Section 3.5 specifies H_L as a vector of vectors (a matrix) of variable length (|x|), yet its length is somehow given as d_h, a simple 1-dimensional vector.
Eq 4 seems to specify that the probability distribution over topics is multiplied together with a probability distribution over words, which should not be possible unless the topics somehow match the vocabulary.

The task in this paper is domain adaptation for generative (auto-regressive) language models. This is a very well researched are with numerous papers over many years. Yet only a couple of very recent papers are cited and compared against. In addition, some of the chosen baselines are not really designed for this task.
Neither prefix-tuning nor DAPT+TAPT were designed or evaluated for language modelling itself, but they are methods for improving downstream performance on a supervised learning task (e.g. text classification).

In Section 3.4 it is mentioned that a bidirectional mask is used, implying that the model is allowed to look into the future. This should not be allowed in a language modelling task.
Section 4.1 is also specifying that the topic vector is averaged over the whole text. It is not clear if this is also done during testing, which would mean that the model gets to see the inputs it needs to generate, again something that would not be allowed.

The first sentence of the abstract claims that the proposed method is making it possible to use language models for unconditional generation. That is what language models already do, without any adaptation. They are conditioned only on the previous context, but so is the proposed model, so the claim is confusing.

Please look into using citet vs citep. The current use of citation formatting is making it difficult to understand the sentence structure.


**Strength And Weaknesses:**

Strengths:
* The of estimating the topic of the text first, then generating based on that is interesting and possibly promising.
* The empirical results are good compared to the chosen baselines.

Weaknesses:
* The clarify of the paper is low, prohibiting understanding of how the method actually works.
* Experiments are performed only on two datasets, both from the same domain. For convincing results regarding the domain adaptation method, more than one domain should be investigated during evaluation.

**Summary Of The Paper:**

The paper investigates domain adaptation for an autoregressive language model.
A latent topic distribution is calculated based on the input words, which is then used to modify the token probabilities during generation.
Evaluation is performed on two review datasets (Amazon and Yelp).

**Summary Of The Review:**

Possibly interesting and useful ideas.
The clarity of the paper needs improvement and the evaluation should be against more relevant baselines and on more datasets.

---

> ### Author Response · Authors · 2022-11-08
> **Confirmation and correction for deep understanding**
>
> Thank you for reviewing our manuscript.
> We submitted a manuscript based on the questions and comments we received,
> where we have highlighted the corrections in red and the references in red underline.
>
> W1) Experiments are performed only on two datasets, both from the same domain. For convincing results regarding the domain adaptation method, more than one domain should be investigated during evaluation.\
> --->The selection of experimental datasets focuses on their suitability for our goal and task, reliability assurance of human evaluation, and reproducibility of these experiments, rather than the number of datasets.
> We need datasets in the similar domain for comparing our framework with domain-adaptive pre-training methods,
> and could not increase the number of available datasets in order to reduce the burden on evaluators, improve consistency of evaluation, and increase reproducibility for other organizations.
>
> C1) For example, Section 3.5 specifies H_L as a vector of vectors (a matrix) of variable length (|x|), yet its length is somehow given as d_h, a simple 1-dimensional vector. \
> --->Apologies for not stating it exactly. We modify is as follows:　$\mathbb{R}^{d_{h}}$-> $\mathbb{R}^{|x| \times d_{h}}$
>
> C2) Eq 4 seems to specify that the probability distribution over topics is multiplied together with a probability distribution over words, which should not be possible unless the topics somehow match the vocabulary. \
> --->Please instruct us with the basis on which you can make such an assertion.
> These topics can be considered as a quantized sample of the underlying word distribution, and be end-to- end learnable together with the model parameters. TSL can update topic related parameters, $W_{Z}$ and $W_{V}$, like $W_{l}^{Q},W_{l}^{K},W_{l}^{V}\in\mathbb{R}^{d_{h}\times d_{k}}$ in Eq (3), including distribution through training (i.e., back propagation with cross validation over tokens and training tasks) like other hidden variable parameters such as $H_{t}$, and have actually updated them.
> Of course, a collapsed Gibbs sampling can also be applied, but at a higher computational cost.
>
> C3) This is a very well researched are with numerous papers over many years. Yet only a couple of very recent papers are cited and compared against. \
> --->We address both the domain gap and catastrophic forgetting by developing model's mechanism, components and objectives, while other studies address each one separately by designing learning methods. Then, we have chosen the most recent and reliable methods among the latter. If there are references to fit this selection perspective and the purpose of their validation, and be cited, please inform us.
>
> C4) In addition, some of the chosen baselines are not really designed for this task. Neither prefix-tuning nor DAPT+TAPT were designed or evaluated for language modelling itself, but they are methods for improving downstream performance on a supervised learning task (e.g. text classification). \
> --->As you pointed out, they are not designed as language modeling, but share the similar issue such the domain adaptation with our framework. Therefore, we prepared data setes and designed experiments to compare their methods with out framework in order to make the best use of them.
>
> C5) In Section 3.4 it is mentioned that a bidirectional mask is used, implying that the model is allowed to look into the future. This should not be allowed in a language modelling task. Section 4.1 is also specifying that the topic vector is averaged over the whole text. It is not clear if this is also done during testing, which would mean that the model gets to see the inputs it needs to generate, again something that would not be allowed. \
> --->Sorry for leaving out the description of one of the variations of our framework.
> As you pointed out, the test phase (text generation) can not look into the future and use a bidirectional mask.
> As topic models treat topics not in word order but in the global information such as word co-occurrence,
> this variant follows this concept and applies the bidirectional mask to only training phase.
> This looks a bit tricky, so we left it out of this paper, but part of it has been left in.
>
> C6) The first sentence of the abstract claims that the proposed method is making it possible to use language models for unconditional generation. That is what language models already do, without any adaptation. They are conditioned only on the previous context, but so is the proposed model, so the claim is confusing. \
> --->As you pointed out, this is obvious and we modify it as follows:
> Our goal is to guide pre-trained language models (PLMs) towards unconditional text generation tasks while resolving the domain gap and avoiding the catastrophic forgetting.

---

> > ### Comment · Reviewer_6wEU · 2022-11-19
> > **Regarding more datasets**
> >
> > "We need datasets in the similar domain for comparing our framework with domain-adaptive pre-training methods, and could not increase the number of available datasets in order to reduce the burden on evaluators, improve consistency of evaluation, and increase reproducibility for other organizations."
> >
> > The task is language modelling, which means no labeling is needed and datasets from different domains are abundant. I understand the concern about human annotation but all except 1 of your evaluation metrics are completely automated. So it should be straightforward to evaluate on many more datasets, even if the human evaluation is performed on a smaller selection of datasets.
> >
> > I would recommend strengthening the evaluation and improving the clarity of the paper further.

---

> > > ### Author Response · Authors · 2022-11-19
> > > **Are our answers insufficient?**
> > >
> > > Thank you for the reply.
> > > It is the domain shift problem that we are trying to solve in this paper, not the presence or absence of labels.\
> > > Of course we could use data across different domains, but that would be unfair to them,
> > > unlike the assumptions of baselines.\
> > > As long as automated evaluation metrics do not completely match human evaluation, we believe that manual evaluation is essential.\
> > > Because we have already applied automated evaluation metrics to these experiments, we would like to know specifically how your comments relate to these experiments.
> > > Our hope is that you will clarify this point for us to take advantage of your comments.
> > >
> > > We have responded to several misconceptions in your review (e.g., C2), did that help you understand?

---

### Decision · Program_Chairs · 2023-01-20

**Decision:**

Reject

**Justification For Why Not Higher Score:**

Please see the meta-review and discussion.

**Justification For Why Not Lower Score:**

Please see the meta-review and discussion.

**Metareview: Summary, Strengths And Weaknesses:**

This paper proposes to adapt pre-trained language models to support unconditional text generation tasks. Reviewers all agreed that this idea of predicting topics first and then generating text is interesting and novel, which can bring good insights and interpretability into the process of language modeling. On the other side, the key concerns in the original reviews after the discussion remain, towards the clarity of this paper, the limited focus of using two datasets from the same domain for evaluation of this domain adaption method, as well as the lack of related work comparison/discussion to other adaption methods.  Reviewers have shared some concrete suggestions that the authors could use to better improve this work.



**Summary Of Ac-Reviewer Meeting:**

N/A